# The Combination of PPARα Agonist GW7647 and Imeglimin Has Potent Effects on High-Glucose-Induced Cellular Biological Responses in Human Retinal Pigment Epithelium Cells

**DOI:** 10.3390/bioengineering12030265

**Published:** 2025-03-08

**Authors:** Nami Nishikiori, Megumi Watanabe, Megumi Higashide, Araya Umetsu, Toshifumi Ogawa, Masato Furuhashi, Hiroshi Ohguro, Tatsuya Sato

**Affiliations:** 1Departments of Ophthalmology, Sapporo Medical University, S1W17, Chuo-ku, Sapporo 060-8556, Japan; nami076@yahoo.co.jp (N.N.); watanabe@sapmed.ac.jp (M.W.); megumi.h@sapmed.ac.jp (M.H.); araya.umetsu@sapmed.ac.jp (A.U.); 2Departments of Cardiovascular, Renal and Metabolic Medicine, Sapporo Medical University, S1W17, Chuo-ku, Sapporo 060-8556, Japan; a08m024@yahoo.co.jp (T.O.); furuhasi@sapmed.ac.jp (M.F.); 3Departments of Cellular Physiology and Signal Transduction, Sapporo Medical University, S1W17, Chuo-ku, Sapporo 060-8556, Japan

**Keywords:** retinal pigment epithelium, metformin, imeglimin, extracellular flux analyzer, ROS

## Abstract

Background: Hyperglycemic changes in the cellular biological properties of retinal pigment epithelium cells are involved in the pathophysiology of diabetic retinopathy (DR). To assess the effects of the new anti-diabetic agent imeglimin (Ime) on DR, the pharmacological effects of Ime and those of metformin (Met) in combination with the PPARα agonist GW7646 (GW) on adult retinal pigment epithelium (ARPE19) cells cultured in high-glucose conditions were compared. Methods: Cell viability, levels of reactive oxygen species (ROS), monolayer barrier function measured by transepit very much helial electrical resistance (TEER), and metabolic functions determined by an extracellular flux analyzer were evaluated. Results: While glucose concentrations did not alter cell viability regardless of the presence of Met or Ime, levels of ROS were significantly increased by the high-glucose conditions, and increased levels of ROS were significantly alleviated by the combination of Ime and GW but not by Met alone. Similarly, TEER values were increased by high-glucose conditions, but the effects of high-glucose conditions were dramatically enhanced by the combination of Ime and GW. Furthermore, a metabolic assay showed that an energetic shift was induced by the combination of Ime and GW, whereas energy status became quiescent with Met or Ime alone. Conclusions: The collective results suggest that Ime in combination with GW has synergetic effects on high-glucose-induced cellular biological changes in ARPE19 cells.

## 1. Introduction

Visual deterioration in patients with diabetes mellitus (DM) is mainly caused by proliferative diabetic retinopathy (PDR) in type 1 DM (T1DM), whereas it is often caused by diabetic macular edema (DME) in patients with type 2 DM (T2DM) [1]. Since the prevalence of T2DM is much higher than that of T1DM, DME is a major cause of visual impairment in patients with DM, and DME is also already present when PDR is found in patients with T2DM [2]. As for the underlying pathogenesis of DME, it has been shown that fluid accumulation in the macular lesion due to vascular leakage caused by the breakdown of blood–retina barriers (BRBs) is primarily involved [3,4]. There are two BRBs, the inner BRB (iBRB) and the outer BRB (oBRB), in which tight junctions between adjacent endothelial or retinal pigment epithelial (RPE) cells are present. In contrast to the simple biological barrier function of the iBRB, which is to maintain the homeostasis of the neural retina during trafficking processes with blood circulation, the oBRB has additional metabolic roles in maintaining photoreceptor functions via the renewal of the photoreceptor outer segment (OS) by the phagocytosis and digestion of shed OS tips [5] and by the transport of major energy sources such as glucose and lactose [6,7], as well as retinoids [8], which are essential for cell survival and a normal visual cycle from the blood circulation to the retina through the RPE layer. In a previous study using a rodent model with DM, a significant breakdown in the oBRB was detected after intravenous injection of fluorescent macromolecules [9]. Furthermore, a recent study using optical coherence tomography (OCT) showed that the thickness of the RPE layer was substantially decreased in patients with DME, suggesting that the thickness of the RPE layer may become an OCT biomarker to detect the early phases of DME [10]. Collectively, the results of studies indicate that RPE cells in the oBRB are involved in the pathogenesis of diabetic retinopathy (DR), including DME, and in fact, in vitro RPE cell models in high-glucose conditions have frequently been used for the study of the pathogenesis of DR and for developing a new therapeutic strategy for DR [11,12,13].

Metformin (Met) is the most widely used oral hypoglycemic agent. It acts by decreasing hepatic gluconeogenesis due to adenosine 5′-monophosphate (AMP)-activated protein kinase (AMPK)-dependent action to facilitate the AMPK-mediated translocation of SLC2A4/glucose transporter type 4 [14,15]. Met also has AMPK-independent actions [16] and is therefore involved in additional and unexpected biological roles in addition to its hypoglycemic action [17]. For instance, it has been shown that the administration of Met inhibits inflammation, generation of reactive oxygen species (ROS), senescence of endothelial cells, programmed cell death, and neovascularization by modulating AMPK-dependent and AMPK-independent actions [18]. In fact, Met has been shown to exert a protective effect on RPE cells in vitro and in vivo under pathogenic conditions [19]. Furthermore, a recent study using proteomics analysis of vitreous humor specimens demonstrated that peroxisome proliferator-activated receptor alpha (PPARα) is a possible upstream regulator of PDR [20], and the activation of PPARα has, in fact, been identified with possible therapeutic targets for DR in human and animal models [21]. Furthermore, co-treatment using a PPARα agonist with Met has been shown to induce better therapeutic effects in various clinical studies for patients with T2DM [22] and for patients with non-alcoholic fatty liver disease (NAFLD) [23] in addition to an in vivo study using an advanced nonalcoholic steatohepatitis (NASH) rodent model [24].

Imeglimin (Ime) is the first agent in a new tetrahydrotriazine class of oral antidiabetic drugs called ‘glimins’, and it is expected to have the potential of solutions for a number of unmet medical needs for patients with T2DM [25,26]. Recent clinical experience with Ime in Japanese and Caucasian patients with T2DM has shown significant and durable antihyperglycemic action with safety and tolerability, as well as a lack of severe hypoglycemi, a in various clinical trials, including trials using combinations with insulin, Met, and other anti-DM drugs [27,28,29]. However, as of the writing of this paper, Ime-induced biological effects on DM and hyperglycemic-related manifestations of DR have not been investigated. Recent studies using diabetic models have shown that Ime also effectively reduces the production of ROS [30,31], which is also the underlying pathogenesis of DR. There has been great interest in elucidating the undetermined therapeutic effects of treatment with Ime, as well as treatment with Ime and a PPARα agonist in combination, on the pathogenesis of DR.

Therefore, to study the effects of Ime and those of Ime and a PPARα agonist in combination on the pathogenesis of DR, we compared the effects of treatment with Ime and of treatment with Ime and the PPARα agonist GW7647 (GW) with the effects of treatment with Met on cellular biological properties, including the generation of ROS, barrier functions, and metabolic functions in the adult RPE cell line ARPE19 under high-glucose conditions.

## 2. Materials and Methods

### 2.1. Cell Culture of Human Adult Retinal Pigment Epithelium 19 Cells (ARPE19 Cells)

After the approval of the current study by the internal review board of Sapporo Medical University (approval number: 312-3190), all experiments using human-derived commercially available ARPE19 cells obtained from the ATCC (#CRL-2302™, date of authentication: 6 June 2018; based on species determination by STR analysis, sterility test, and human pathogenic virus test, in addition to common cellular characterization including morphology, growth, and viability, as attached in the Appendix A) were carried out in compliance with the tenets of the Declaration of Helsinki. ARPE19 cells were cultured in 150 mm planar two-dimensional (2D) culture dishes until 90% confluence under standard normoxia conditions (37 °C, 5% CO_2_) in a culture medium consisting of LG-DMEM (5.5 mM glucose) supplemented with 10% FBS, 1% L-glutamine, and 1% antibiotic–antimycotic and were further maintained by daily changes of the medium. For the evaluation of the effects of high glucose (50 mM) not supplemented with or supplemented with various combinations of Ime, Met, and GW, 2D cultured ARPE19 cells were incubated with high glucose and those combinations for 24 h until the measurements described below.

### 2.2. Cell Viability Test

A commercially available kit (Cell Counting Kit-8, Dojindo, Tokyo, Japan) was used to determine the viability of ARPE19 cells, as described in our previous report [32].

### 2.3. Measurement of Levels of Reactive Oxygen Species (ROS)

To characterize the cellular function of ARPE19 cells with various combinations of Ime, Met and GW, levels of ROS were measured as described in our previous report [33].

### 2.4. Monolayer Barrier Function of ARPE19 Cells Assessed by Transepithelial Electrical Resistance (TEER)

To evaluate the monolayer barrier function of ARPE19 cells with various combinations of Ime, Met and GW, measurements of TEER values were carried out as described previously [34,35]. The morphology of the monolayers was observed using a phase-contrast microscope (PC, Nikon ECLIPSE TS2; Tokyo, Japan).

### 2.5. Seahorse Cellular Metabolic Function Measurement

After the treatment or non-treatment of planar cultured ARPE19 cells with various combinations of Ime, Met and GW, measurements of oxygen consumption rate (OCR) and extracellular acidification rate (ECAR) were carried out using a Seahorse XFe96 Bioanalyzer (Agilent Technologies, Santa Clara CA, USA) as described in our previous report [32]. Calculations of metabolic indices used were also described in our previous report [32].

### 2.6. Other Analytical Methods

Total RNA extraction from the various 2D cultured cells and subsequent reverse transcription and quantitative real-time PCR (qRT-PCR) were carried out as previously reported [36,37] using specific primers and probes (Appendix A). All statistical analyses were carried out using Graph Pad Prism 9 or 10 (GraphPad Software, San Diego, CA, USA) as described in a recent study [36,37].

## 3. Results

### 3.1. Effects of Concentrations of Met and Ime on Cell Viability in ARPE19 Cells Under Low- and High-Glucose Conditions 

To study the effects of antidiabetic agents including Met and Ime in the presence or absence of the PPARα agonist GW7647 (GW) on DR-related oxidative stress of the RPE, high-glucose-stimulated ARPE19 cells were used as a model of the damaged RPE, mimicking DM-induced oxidative stress. Prior to evaluation of Met- or Ime-induced effects, the viability of ARPE19 cells in the presence of various concentrations (0 to 5 mM) of these agents was measured to confirm that the cytotoxic effects induced by the administered drugs were at negligible levels. As shown in Figure 1, (1) no significant cytotoxic effects of Met or Ime were detected in concentration ranges between 0 and 2 mM on day 1 under low-glucose and high-glucose conditions (panel A), and (2) a toxic effect was observed at a 5 mM concentration of these reagents on day 1 and upon more than 2 days’ treatment of the cell cultures at 2 mM of these reagents under high-glucose conditions (panel B). Therefore, we used 2 mM of Met and Ime on day 1 for the following experiments. Alternatively, 10 μM of GW was used according to a previous study [38] in addition to the manufacturer’s recommendation.

### 3.2. Effects of Met, Ime, and/or GW7647 on Levels of Reactive Oxygen Species (ROS) in ARPE19 Cells

To evaluate the effects of high-glucose stimulation on oxidative stress in ARPE 19 cells, levels of ROS were measured, and it was found that they were significantly increased in high-glucose conditions (Figure 2A). As shown in Figure 2B, the levels of ROS in high-glucose conditions were slightly and substantially decreased by Met and Ime, respectively, and they were also significantly decreased by monotreatment with GW. Furthermore, the addition of GW enhanced the Met- or Ime-induced effects, suggesting that the PPARα agonist GW has synergetic effects with Met and Ime on high-glucose-induced oxidative stress.

### 3.3. Effects of Met, Ime, and/or GW7647 on Transendothelial Electrical Resistance (TEER) Values of 2D ARPE19 Cell Monolayers Under Low- and High-Glucose Conditions

To further study the effects of Met, Ime, and/or GW on oxidative stress-induced biological changes in the RPE, the barrier function of RPE cells serving as a putative outer blood–retina barrier (oBRB) was evaluated using TEER measurement of ARPE19 cell monolayers. As shown in Figure 3, the TEER values of 2D ARPE19 cell monolayers were significantly increased under high-glucose conditions compared with those under low-glucose conditions, and the values were further increased by monotreatment with Met, Ime, or GW. The addition of GW caused further enhancement of the Met- or Ime-induced increase in TEER values.

These effects in TEER experiments were supported by phase-contrast images of AR-PE19 cell monolayers under various conditions, showing that the density of cells under high-glucose conditions was greater than that under low-glucose conditions, and the density of cells was further and synergistically enhanced by an antidiabetic agent and GW (Figure 4). Collectively, the results indicated that the antidiabetic agents Met and Ime solely reinforced the barrier functions of the RPE and that their effects were further enhanced by high-glucose conditions. The addition of the PPARα agonist GW may be a protective phenomenon by suppressing DM-related oxidative stress, as shown in Figure 2.

As possible underlying molecular mechanisms causing these changes in barrier functions of the RPE, it was speculated that different expression levels of tight-junction-related molecules or different levels of cell proliferation may be involved. As shown in Figure 5, although the qRT-PCR analysis of major tight-junction-related molecules, ZO-1, Occludin, connexin 43 (Cx43), and Claudin-1, in the RPE showed that the mRNA expression of these molecules fluctuated differently among the conditions, those changes did not correspond to the changes in TEER values, as shown below.

In contrast, the proliferation of ARPE19 cells was substantially increased under high-glucose conditions compared to that under low-glucose conditions, and the proliferation of the cells under high-glucose conditions was further enhanced by Ime or GW (Figure 6), suggesting that the physical property of ARPE19 monolayers may be modulated by diverse cell proliferations affected by different glucose conditions, antidiabetic agents, and/or GW.

### 3.4. Effects of Metformin, Imeglimin, and/or GW7647 on Cellular Metabolic Functions in ARPE19 Cells

Finally, to observe the effects of Met, Ime, and/or GW on cellular metabolism, we evaluated metabolic indices using an extracellular flux analyzer (Figure 7). First, ARPE19 cells cultured under high-glucose (50 mM) conditions showed a trend of increased proton leak, which could reflect increased ROS, compared to those cultured under low-glucose (5.5 mM) condition, but the difference did not reach statistical significance. In addition, there were no significant differences in mitochondrial function or glycolytic functions between the two conditions. Next, the effects of various agents were evaluated in ARPE19 cells cultured under high-glucose conditions. Treatment with GW shifted cellular metabolism energetically, whereas Met or Ime alone shifted cellular metabolism from mitochondrial respiration to glycolysis. Surprisingly, the combination of Ime and GW increased ATP-linked respiration and mitochondrial maximal respiration without a decrease in proton leak or spare respiratory capacity. Such an effect was not observed with the combination of Met and GW. Collectively, the results indicated that the effects of Met or Ime alone on cellular metabolism were similar, but a distinctive interaction with GW was observed only for Ime in ARPE19 cells cultured under high-glucose conditions.

## 4. Discussion

Several studies have revealed that chronic hyperglycemia induces unfavorable complications, such as inflammation [39], as well as an accumulation of intracellular ROS, thereby leading to cellular damage. ROS are known to be produced intracellularly by impaired mitochondrial functions or by interactions with exogenous sources. In the pathogenesis of DR, it is known that the oxidative stress-induced elevation of the levels of ROS plays a critical role in the development of DR [40]. Hyperglycemia is known to induce the generation of ROS in cells [41] and, in fact, elevated levels of ROS in retinal cells were reported in diabetic rats compared to the levels in retinal cells in nondiabetic rats [42].

Ime is a recently launched drug in a new class of antidiabetic drugs that structurally and functionally resembles Met, although the pharmacological properties of Ime have not been fully determined yet. A recent study using an extracellular flux analyzer and RNA-sequencing analysis showed that Ime reduced the oxygen consumption rate coupled to ATP production of HepG2 cells or mouse primary hepatocytes by activating AMPK, similarly to Met, though the potency of the effect of Ime was less than that of Met [43]. Although Ime- and Met-induced gene expression profiles in HepG2 cells were overall similar, an up-regulation of genes encoding proteins of mitochondrial respiratory complex III and complex I was observed with Ime but not with Met [43]. Therefore, the results of that study suggested that Ime and Met had similar pharmacological activities in cultured hepatocytes, whereas both drugs may exert different effects on mitochondrial function [43]. In fact, although the mechanism of Ime-induced suppression of mitochondrial function has not been elucidated, it was shown that Ime did not inhibit the activities of complex I or complex V in hepatocytes, whereas Met inhibited the activities of both complexes [44]. Furthermore, although the effects of both drugs on mitochondrial respiration and AMPK activation were overall similar, the potency of the effect of Met was greater than that of Ime [43]. In addition, such differences between Ime and Met were more apparent in primary hepatocytes than in HepG2 cells [43], which is consistent with the results of a previous study showing that the effects of Ime on the ATP/ADP ratio and glucose production in primary hepatocytes were less potent than the effects of Met [44]. In contrast to such superiority of the effect of Met on mitochondrial respiration in hepatocyte function compared to that of Ime, our study showed that Ime and Met had similar effects on decreasing elevated levels of ROS (Figure 2) and increasing TEER values (Figure 3) in ARPE19 cells induced by high glucose, though the effects of Ime were more potent than those of Met. In addition to levels of ROS and TEER values, Ime, but not Met, had additional synergistical effects with the PPARα agonist GW on mitochondrial and glycolytic functions (Figure 6). The results indicated that metabolic activation by GW can be observed in the presence of Ime, but not in the presence of Met, being consistent with the results of a previous study showing that Met shows uncompetitive inhibition of the respiratory chain, whereas Ime shows competitive inhibition of the respiratory chain [44]. Therefore, these collective observations suggested that (1) more potent effects of Ime than those of Met on oxidative stress and barrier function of the oBRB and cellular metabolic functions may effectively inhibit the development of DR and (2) different pharmacological effects of Ime and Met may be exclusively tissue- and cell-specific or depend on different glucose conditions. In support of this, in a recent study in rats showed that orally administered Ime was detected in the brain by passing through the blood–brain barrier (BBB) [45], suggesting that orally administered Ime may also be intraocularly delivered. Furthermore, it was shown that Ime positively affected high-glucose-induced mitochondrial dysfunction and ROS production, which are involved in the pathogenesis of neurodegenerative diseases [46], and thus, Ime prevented cell death during oxidative stress due to hyperglycemia [46,47].

Since insignificant long-term side effects of Met and Ime in clinical settings have been confirmed, these agents are already used as common therapy for patients with hyperglycemia [14,15,25,26]. However, in the present in vitro study, treatment with 2 mM Met and 2 mM Ime did not affect the cellular viability of ARPE19 cells for up to 24 h, whereas more than 2 days’ culture with these agents slightly but significantly decreased cellular viability without additional effects for up to 5 days (Figure 1). These findings may appear to contradict the effect of Met and Ime on reducing ROS production (Figure 2). Considering the results of the Seahorse analysis showing that 2 mM Met and 2 mM Ime shifted the cellular metabolism of ARPE19 from mitochondrial respiration to glycolysis (Figure 7), and the fact that retinal pigment epithelial cells are vulnerable to hypoxia or nutrition deprivation, the effects of Met and Ime on cell viability are presumably due to cell metabolic deprivation rather than ROS-dependent cellular damage or cell death. From this point of view, the fact that metabolic activation by GW was clearly observed when it was used in combination with Ime, but not with Met, supports the notion that Met and Ime differently affect cellular biological properties including redox status and cellular metabolism. Collectively, these findings suggest that the effect of the Ime-induced reduction in levels of ROS on cellular metabolism may be minimized by using Ime in combination with PPARα agonists such as GW. To confirm our speculation, further studies using in vivo animal models or clinical studies focusing on the long-term effects on DR in Met and Ime in combination with PPARα agonists are warranted.

We acknowledge that there are several limitations to the current study. Firstly, we only performed an analysis of the ARPE19 cell line. Secondly, the mechanisms by which the synergistic effects of Ime with the PPARα agonist GW7647, especially on the barrier function of ARPE19 cell monolayers, are induced have not been elucidated. Thirdly, GW7647 is not clinically used despite the fact that fibrates such as fenofibrate, bezafibrate, and pemafibrate are currently used in patients. Fourthly, the investigation of the effects of different concentrations of the drugs, as well as different durations of treatment, including long-term treatment, is important for a better understanding of the efficacies of the drugs. Recently, it was shown that AMPK activities in HepG2 cells and mouse primary hepatocytes were increased by Ime and Met in a dose-dependent manner with doses ranging from 0.25 to 10 mM [43], and 1 mM concentrations of both drugs were used to study glucose production, the ATP/ADP ratio, and mitochondrial function in rat hepatocytes [44]. Even if different cell types were considered, these observations suggested that the concentration of 2 mM of Ime and Met used in our study may be a suitable experimental condition for evaluating the effects of the drugs. Furthermore, in those studies, the effects of the drugs were evaluated in much shorted exposure time periods, such as 0.5 to 12 h [43], than the exposure period of 24 h in our study. In contrast to in vivo experiments using animal models, since cell toxicity effects should be carefully considered in the case of in vitro cell culture experiments, insignificant levels of cytotoxic effects by both drugs until 24 h were confirmed, as shown in Figure 1. Fifthly, while monotreatment with Ime and treatment with Met or Ime in combination with GW led to a significant decrease in ROS under high-glucose conditions, these levels were much lower than those in low-glucose conditions, as shown in Figure 2. Although the biological significance of such decreased levels of ROS after treatment, which were under the levels of ROS at physiological low-glucose conditions (5.5 mM), remains to be elucidated, this may bring therapeutic benefits in diabetic and none-diabetic RPE. Therefore, to overcome these study limitations, additional study is required to determine the pharmacological effects of Ime on the pathogenesis of DR using additional intraocularly derived cells, including primary cultured RPE cells and in vivo animal models, as well as using clinically used PPARα agonists in our next projects.

## 5. Conclusions

Ime showed stronger effects than Met, along with additional synergistical effects with GW, on reducing oxidative stress and improving barrier function in ARPE19 cells under high-glucose conditions, suggesting the potential of Ime for inhibiting DR progression. These pharmacological effects were observed even under high-glucose conditions, suggesting that they can be independent of glycemic management in patients with DM. Clinical trials are needed to verify the efficacy of Ime against DR at various stages in the future.

## Figures and Tables

**Figure 1 bioengineering-12-00265-f001:**
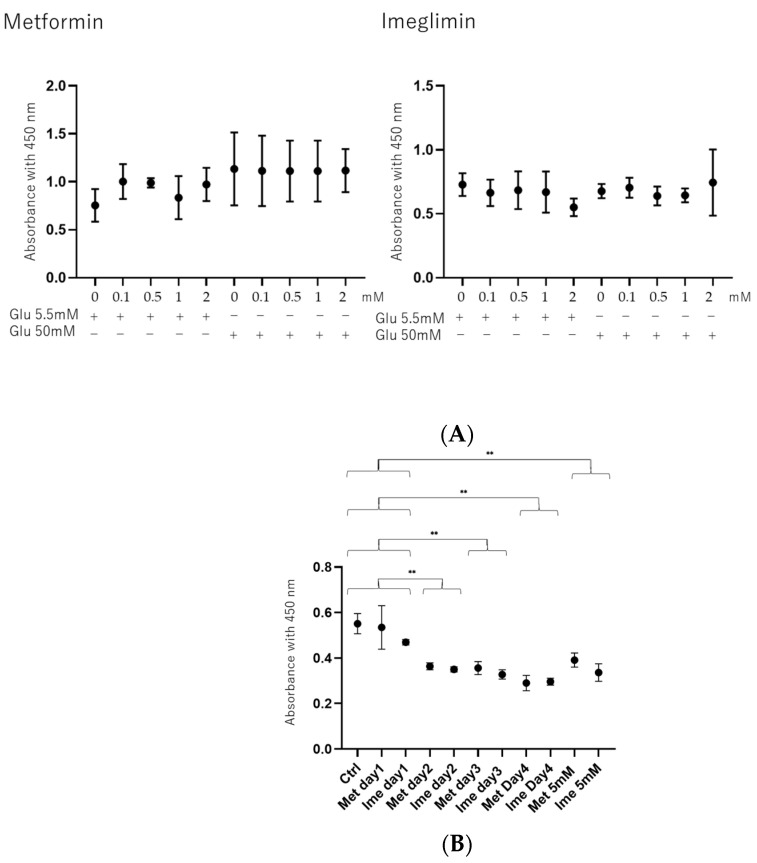
**Cytotoxic effects of Met and Ime on 2D cultured ARPE 19 cells**. (Panel **A**): Cytotoxic effects of different concentrations of metformin (Met) and imeglimin (Ime) under low-glucose conditions (5.5 mM) and high-glucose conditions (50 mM) on ARPE19 cells on day 1. (Panel **B**): Cytotoxic effects of 2 or 5 mM of Met and Ime under high-glucose conditions (50 mM) on ARPE19 cells on day 1 through day 4. Living ARPE19 cells detected using a Cell Counting Kit-8 (CCK-8) were plotted (n = 8). Met: metformin; Ime: imeglimin. ** *p* < 0.01.

**Figure 2 bioengineering-12-00265-f002:**
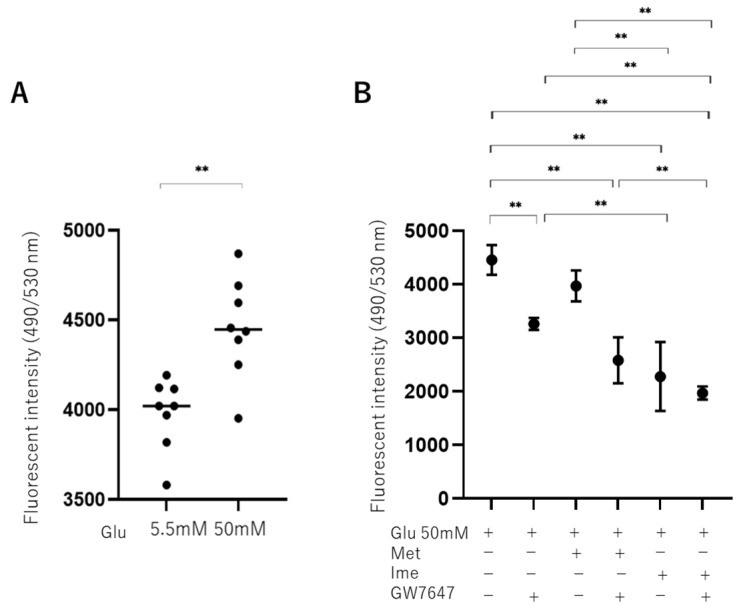
**Effects of Met, Ime, and/or GW7647 on levels of reactive oxygen species (ROS) in ARPE19 cells under different glucose conditions.** Planar ARPE19 cultured cells were prepared under low-glucose (5.5 mM) conditions for 3 days. The cells were further cultured for 24 h (control) or were further cultured under high-glucose (50 mM) conditions for 24 h in the absence or presence of 2 mM metformin (Met), 2 mM imeglimin (Ime), and/or 10 μM GW7647 (GW). The ARPE19 cells were then subjected to measurement of reactive oxygen species (ROS), and the values were plotted. ROS levels are compared between low- and high-glucose conditions (Panel **A**). ROS levels are measured under high-glucose conditions in the presence or absence of Met and Ime, with or without GW7647 (Panel **B**). Triplicate experiments were performed using fresh preparations. ** *p* < 0.01.

**Figure 3 bioengineering-12-00265-f003:**
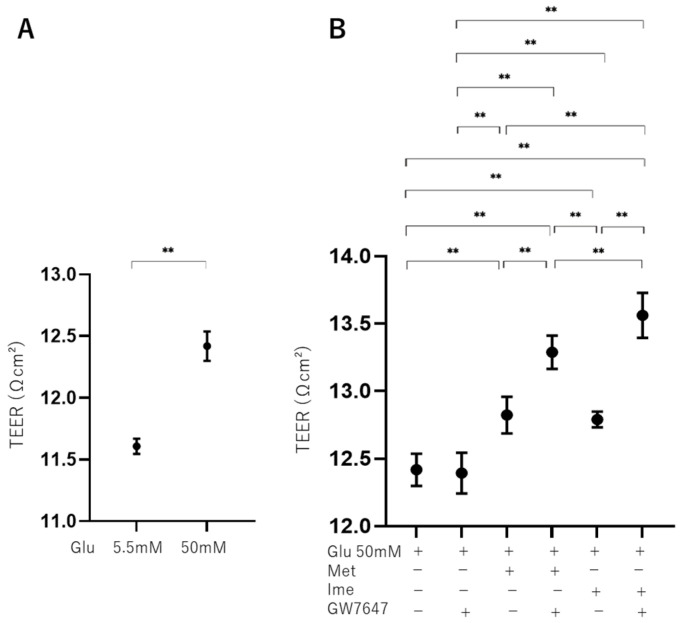
**Effects of Met, Ime, and/or GW7647 on monolayer barrier function of ARPE19 cells assessed by transendothelial electrical resistance (TEER) under different glucose conditions.** Planar ARPE19 cell monolayers were prepared under low-glucose (5.5 mM) conditions for 3 days. The cells were further cultured for 24 h (control) or were further cultured under high-glucose (50 mM) conditions for 24 h in the absence or presence of 2 mM metformin (Met), 2 mM imeglimin (Ime), and/or 10 μM GW7647 (GW). As monolayer barrier function, TEER values were determined by electric resistance (Ωcm^2^) measurements and plotted. TEER values are compared between low- and high-glucose conditions (Panel **A**). TEER values are measured under high-glucose conditions in the presence or absence of Met and Ime, with or without GW7647 (Panel **B**). Triplicate experiments were performed using fresh preparations. ** *p* < 0.01.

**Figure 4 bioengineering-12-00265-f004:**
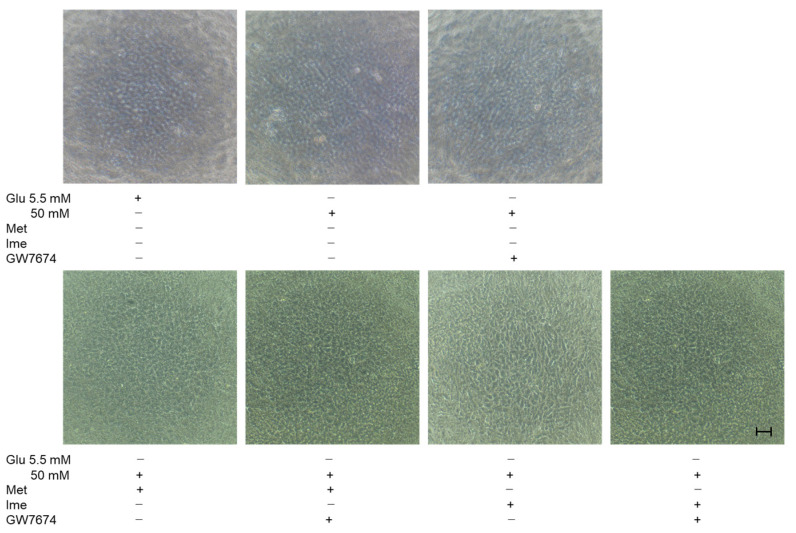
**Representative phase-contrast microscopy images of ARPE19 cell monolayers under various conditions.** Planar ARPE19 cell monolayers were prepared under low-glucose (5.5 mM) conditions for 3 days. The cells were further cultured for 24 h (control) or were further cultured under high-glucose (50 mM) conditions for 24 h in the absence or presence of 2 mM metformin (Met), 2 mM imeglimin (Ime), and/or 10 μM GW7647 (GW). Representative phase-contrast images of ARPE19 cell monolayers are shown. Scale bar: 100 μm. Duplicated experiments were performed using fresh preparations. (n = 3 each).

**Figure 5 bioengineering-12-00265-f005:**
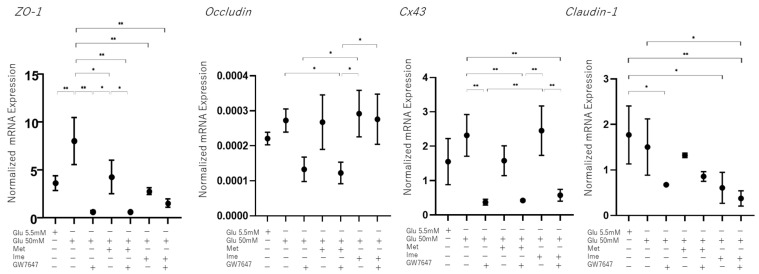
**Effects of Met, Ime, and/or GW7647 on *Zo-1,*
*Occludin, Cx43, Claudin-1* expression in 2D cultured ARPE19 cells under different glucose conditions**. Planar cultured ARPE19 cells were prepared under low-glucose (5.5 mM) conditions for 3 days. The cells were further cultured for 24 h (control) or were further cultured under high-glucose (50 mM) conditions for 24 h in the absence or presence of 2 mM metformin (Met), 2 mM imeglimin (Ime), and/or 10 μM GW7647 (GW). By performing qPCR analysis on each sample, the mRNA expression levdels of major tight-junction-related components, *Zo-1, Occludin, Cx43* and *Claudin-1* were evaluated. Duplicated experiments were performed using fresh preparations (n = 5 each). * *p* < 0.05, ** *p* < 0.01.

**Figure 6 bioengineering-12-00265-f006:**
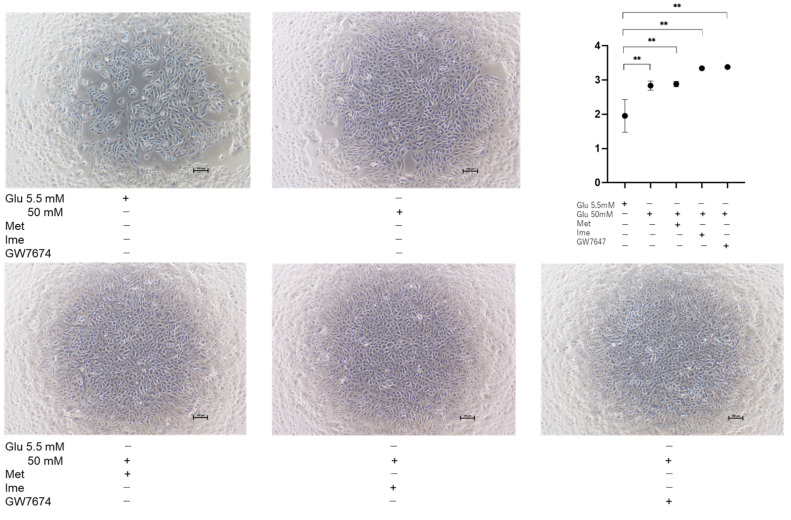
**Proliferation of ARPE19 cells under various conditions**. Planar ARPE19 cells were prepared under low-glucose (5.5 mM) conditions for 3 days. The cells were further cultured for 24 h (control) or were further cultured under high-glucose (50 mM) conditions for 24 h in the absence or presence of 2 mM metformin (Met), 2 mM imeglimin (Ime), or 10 μM GW7647 (GW). Representative phase-contrast images of ARPE19 cells and plots of cell densities are shown. Scale bar: 100 μm. Duplicated experiments were performed using fresh preparations (n = 3 each). ** *p* < 0.01.

**Figure 7 bioengineering-12-00265-f007:**
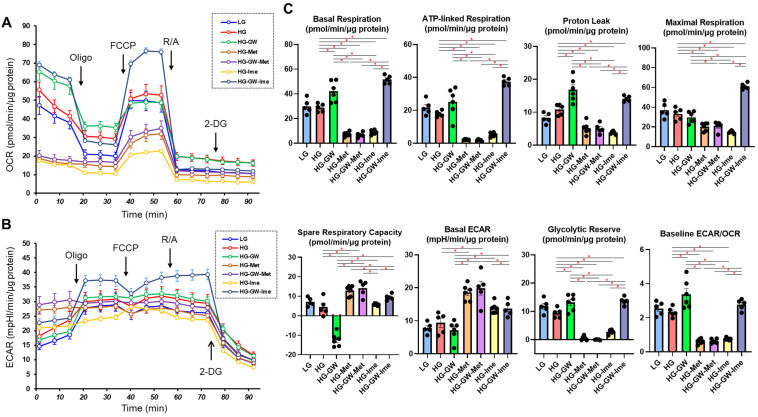
**Effects of Met, Ime, and/or GW7647 on Seahorse cellular metabolic functions of ARPE19 cells.** Planar ARPE19 cells were prepared under a low- glucose (5.5 mM) conditions for 3 days. The cells were further cultured for 24 h (control) or were further cultured under a high- glucose (50 mM) conditions for 24 h in the absence or presence of 2 mM metformin (Met), 2 mM imeglimin (Ime), and/or 10 μM GW7647 (GW). Each specimen was then subjected to Seahorse real-time metabolic function analysis. Plots of OCR values (Panel **A**), plots of ECAR values (Panel **B**) and key metabolic indices (Panels **C**). All experiments were carried out using fresh preparations (n = 5–6). * *p* < 0.05. Since the statistical difference between each agent group and the LG group was the same as the difference between the HG group, the figure does not include an asterisk mark to indicate statistical significance.

## Data Availability

The original contributions presented in this study are included in the article/Appendix A; further inquiries can be directed to the corresponding author.

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
