# Peer review of "The Combination of PPARα Agonist GW7647 and Imeglimin Has Potent Effects on High-Glucose-Induced Cellular Biological Responses in Human Retinal Pigment Epithelium Cells"

_bioengineering, 2025, doi:10.3390/bioengineering12030265_

Round 1
Reviewer 1 Report
Comments and Suggestions for Authors
In this work, the authors highlight the need for study of Combination of the PPARα agonist GW7646 and imeglimin has potent effects on high glucose-induced cellular biological responses in human retinal pigment epithelium cells. The manuscript is mostly well written. I have some major comments and suggestion(s) below.
11. There is a plagiarism which should be minimized to 10%. Especially, in methodology section. The sections 2.1 to 2.6 seem to be almost plagiarized, which might affect the credibility of manuscript. Although, it seems the authors previous works methodology has been adapted. This need to be revised carefully to maintain the credibility of the present study.
22. The abstract's phrasing appears to be inconsistent; it may be reorganized so that readers understand it.
33. In Methodology Section 2.1 Line 98-99: Please provide the approval number by the internal review board of Sapporo Medical University.
44. For Section 2.1.: Give specifics on cell line authentication, such as the date and time of the authentication and the methods that were employed.
55. Discussion is too short should provide summative assessment of present findings.
66. When submitting revisions, all figures must be of high-quality, and the original source images must be provided. For clarity and improved comprehension, a copy of Figure 5 should be added in supplementary file with higher resolution.
77. I encourage authors to include a separate section following the conclusion on Future perspectives of hyperglycemia-induced mitochondrial dysfunction on studies outcome considering production of ROS.

Author Response
Dear Editor,
Thank you very much for the constructive comments concerning our manuscript “Combination of the PPARα agonist GW7646 and imeglimin has potent effects on high glucose-induced cellular biological responses in human retinal pigment epithelium cells.”. We carefully checked all of the Editor and reviewers’ comments and prepared a revised version of our paper that takes these comments into account. The changes are listed below.
Editors’ comment
As pointed out, plagiarism is reduced.
Reviewer 1 comments
In this work, the authors highlight the need for study of Combination of the PPARα agonist GW7646 and imeglimin has potent effects on high glucose-induced cellular biological responses in human retinal pigment epithelium cells. The manuscript is mostly well written. I have some major comments and suggestion(s) below.
- There is a plagiarism which should be minimized to 10%. Especially, in methodology section. The sections 2.1 to 2.6 seem to be almost plagiarized, which might affect the credibility of manuscript. Although, it seems the authors previous works methodology has been adapted. This need to be revised carefully to maintain the credibility of the present study.
Answer; We sincerely appreciate your excellent comment. As suggested, plagiarism is reduced.
- The abstract's phrasing appears to be inconsistent; it may be reorganized so that readers understand it.
Answer; We sincerely appreciate your excellent comment. As pointed out, the abstract is changed to structured form as follow by instruction of Bioengineering.
- In Methodology Section 2.1 Line 98-99: Please provide the approval number by the internal review board of Sapporo Medical University.
Answer; We sincerely appreciate your excellent comment. As pointed out, the approval number (approved number, 312-3190) by the internal review board of Sapporo Medical University is included.
- For Section 2.1.: Give specifics on cell line authentication, such as the date and time of the authentication and the methods that were employed.
Answer; We sincerely appreciate your excellent comment. As pointed out, information related to specifics on cell line authentication is included and PDF of certification from ATCC is also attached in supplemental material: ‘the date of the authentication: June 6, 2018 based on species determination by STR analysis, sterility test, human pathogenic virus test in addition to common cellular characterization including morphology, growth and viability as attached in a supplemental material.’.
- Discussion is too short should provide summative assessment of present findings.
Answer; We sincerely appreciate your excellent comment. As pointed out, summative assessment of present findings was ambiguous and therefore, 2nd paragraph of discussion was rewritten: ‘Ime is a recently launched drug in a new class of antidiabetic drugs that structurally and functionally resembles Met, although the pharmacological properties of Ime have not been fully determined yet. A recent study using an extracellular flux analyzer and RNA-sequencing analysis has shown that Ime reduced the oxygen consumption rate coupled to ATP production of HepG2 cells or mouse primary hepatocytes by activating AMPK similarly to Met, though the potency of the effect of Ime was less that that of Met [42]. Although Ime- and Met-induced gene expression profiles in HepG2 cells were overall similar, up-regulation of genes encoding proteins of mitochondrial respiratory complex III and complex I was observed with Ime but not with Met [42]. Therefore, the results of that study suggested that Ime and Met had similar pharmacological activities in cultured hepatocytes, whereas both drugs may exert different effects on mitochondrial function [42]. In fact, although the mechanism of Ime-induced suppression of mitochondrial function has not been elucidated, it was shown that Ime did not inhibit the activities of complex I or complex V in hepatocytes, whereas Met inhibited the activities of both complexes [43]. Furthermore, although the effects of both drugs on mitochondrial respiration and AMPK activation were overall similar, the potency of the effect of Met was greater than that of Ime [42]. In addition, such differences between Ime and Met were more apparent in primary hepatocytes than in HepG2 cells [42], being consistent with the resultes of a previous study showing that the effects of Ime on ATP/ADP ratio and glucose production in primary hepatocytes were less potent than the effects of Met [43]. In contrast to such superiority of the effect of Met compared to that of Ime on mitochondrial respiration in hepatocyte function, our study showed that Ime and Met had similar effects in decreasing elevated levels of ROS (Fig. 2) and increasing TEER values (Fig. 3) of ARPE19 cells induced by high glucose, though the effects of Ime were more potent than those of Met. In addition to levels of ROS and TEER values, Ime, but not Met, had additional synergistical effects with the PPAR agonist GW on mitochondrial and glycolytic functions (Fig. 6). The results indicated that metabolic activation by GW can be observed in the presence of Ime, but not in the presence of Met, being consistent with the results of a previous study showing that Met shows uncompetitive inhibition of the respiratory chain, whereas Ime shows competitive inhibition of the respiratory chain [43]. Furthermore, these findings also reflect the fact that the risk of side effects of lactic acidosis is lower with Ime than that with Met [44]. Therefore, these collective observations suggested that 1) more potent effects of Ime than those of Met on oxidative stress and barrier function of the oBRB and cellular metabolic functions may effectively inhibit the development of DR and 2) different pharmacological effects of Ime and Met may exclusively be tissue and cell-specific or depend on different glucose conditions. In support of this, in a recent study in rats, showed that orally administered Ime was detected in the brain by passing through the blood–brain barrier (BBB) [45], suggesting that orally administered Ime may also be intraocularly delivered. Furthermore, it was shown that Ime positively affected high glucose-induced mitochondrial dysfunction and ROS production, which are involved in the pathogenesis of neurodegenerative diseases [46], and thus Ime prevented cell death during oxidative stress due to hyperglycaemia [46,47].’.
- When submitting revisions, all figures must be of high-quality, and the original source images must be provided. For clarity and improved comprehension, a copy of Figure 5 should be added in supplementary file with higher resolution.
Answer; We sincerely appreciate your excellent comment. As pointed out, all figures are improved as much as we can, and a copy of Figure 5 is attached in supplementary file with higher resolution. In addition, using the original source images, new figures showing morphology of cell monolayer for TEER and cell proliferation is included.
- I encourage authors to include a separate section following the conclusion on Future perspectives of hyperglycemia-induced mitochondrial dysfunction on studies outcome considering production of ROS.
Answer; We sincerely appreciate your excellent comment. As suggested, conclusion is changed:’ Ime showed stronger effects than those of Met, along with additional synergistical effects with GW, in reducing oxidative stress and improving barrier function under a high glucose condition in ARPE19 cells, suggesting the potential of Ime for inhibiting DR progression. These pharmacological effects were observed even under a high glucose condition, suggesting that they can be independent of glycemic management in patients with DM. Clinical trials are needed to verify the efficacy of Ime against DR at various stages in the future.’.
Reviewer 2 Report
Comments and Suggestions for Authors
This MS entitled “Combination of the PPARα agonist GW7646 and imeglimin has two potent effects on high glucose-induced cellular biological responses in human retinal pigmented epithelial cells” presented a study that analyzed the effects of three substances being already used or proposed to be used for the treatment of diabetes mellitus-related retinopathy. The experimental approach consists of an in vitro test for the short-time effect (24h) of low versus high glucose conditions and different drug treatments on a cell line of human retinal pigmented epithelial cells (ARPE19). The tested drugs were imeglimin (Ime) and metformin (Met) in combination with the PPARα agonist GW7646 (GW). The experimental readouts showed significant differences between treatments as regarding ROS levels, transepithelial electrical resistance (TEER), expression of several membranar molecules, and metabolic values of cellular respiration and glycolysis.
While all the treatments lead to a significant decrease in ROS, these levels were much lower than in low glucose conditions, a result that challenges their therapeutic benefit.
Looking at the changes of the cellular barrier function using the RNA-level expression of several membrane molecules as readouts, it is not clear how the barrier function was modified. Some expression levels for the markers were lower than in low glucose treatments, and some were higher. These experiments need further explanations and a complementary approach at the protein level.
Again, the TEER values for the treatment with Met and Ime were significantly increased compared to the high glucose condition and even higher than in a low glucose condition. This effect needs further explanation, addressing the possible toxic effects vesus therapeutic benefits.
Very interestingly, the metabolic assays showed that Ime and Met induced an energetic shift towards glycolysis, while GW increased cellular respiration. This aspect is very interesting and could be explored to explain the side effects of Ime and Met in clinical studies.
In order to be relevant, this MS should address different concentrations of the drugs as well as different time points, including long-term treatments. The questions generated by the experimental results should be discussed accordingly, and the cell toxicity effects should be carefully considered. These answers could indeed be important for the advancement of treatments for diabetes-related retinopathy.
Author Response
Dear Editor,
Thank you very much for the constructive comments concerning our manuscript “Combination of the PPARα agonist GW7646 and imeglimin has potent effects on high glucose-induced cellular biological responses in human retinal pigment epithelium cells.”. We carefully checked all of the Editor and reviewers’ comments and prepared a revised version of our paper that takes these comments into account. The changes are listed below.
Editors’ comment
As pointed out, plagiarism is reduced.
Reviewer 2 comments
This MS entitled “Combination of the PPARα agonist GW7646 and imeglimin has two potent effects on high glucose-induced cellular biological responses in human retinal pigmented epithelial cells” presented a study that analyzed the effects of three substances being already used or proposed to be used for the treatment of diabetes mellitus-related retinopathy. The experimental approach consists of an in vitro test for the short-time effect (24h) of low versus high glucose conditions and different drug treatments on a cell line of human retinal pigmented epithelial cells (ARPE19). The tested drugs were imeglimin (Ime) and metformin (Met) in combination with the PPARα agonist GW7646 (GW). The experimental readouts showed significant differences between treatments as regarding ROS levels, transepithelial electrical resistance (TEER), expression of several membranar molecules, and metabolic values of cellular respiration and glycolysis. While all the treatments lead to a significant decrease in ROS, these levels were much lower than in low glucose conditions, a result that challenges their therapeutic benefit.
- Looking at the changes of the cellular barrier function using the RNA-level expression of several membrane molecules as readouts, it is not clear how the barrier function was modified. Some expression levels for the markers were lower than in low glucose treatments, and some were higher. These experiments need further explanations and a complementary approach at the protein level.
- Again, the TEER values for the treatment with Met and Ime were significantly increased compared to the high glucose condition and even higher than in a low glucose condition. This effect needs further explanation, addressing the possible toxic effects vesus therapeutic benefits.
Answer for comment #2 and 3; We sincerely appreciate your excellent comment. mRNA expression of selected tight junction proteins fluctuated among various experimental conditions. As pointed out, I totally agree that to elucidate unidentified underlying mechanism for TEER results in addition to diverse effects of Ime and Met, results of our challenging experiment may unfortunately be difficult to read clear explanation because additional factors should be involved. Instead, to convince our TEER results, additional phase contrast images of ARPE19 cell monolayers and cell proliferation to support difference of TEER values among different conditions are included. In addition, those informations are included in corresponding results; ‘To further study the effects of Met, Ime and/or GW on oxidative stress-induced biological changes in the RPE, the barrier function of RPE cells serving as a putative outer blood-retinal barrier (oBRB) was evaluated using TEER measurement of ARPE19 cell monolayers. As shown in Fig. 3, TEER values of 2D ARPE19 cell monolayers were significantly increased under a high glucose condition compared with those under a low glucose condition and the values were further increased by monotreatment of Met, Ime or GW. The addition of GW caused further enhancement of Met- or Ime-induced increase in the TEER values. These effects in TEER experiments were supported by phase contrast images of ARPE19 cell monolayers under various conditions showing that the density of cells under a high glucose condition was greater than that under a low glucose condition and the density of cells was further and synergistically enhanced by an antidiabetic agent and GW (Fig. 4). Collectively, the results indicated that the antidiabetic agents Met and Ime solely reinforced barrier functions of the RPE and that their effects were further enhanced by a high glucose condition. The addition of the PPARα agonist GW may be a protective phenomenon by suppressing DM-related oxidative stress as shown in Fig. 2. As possible underlying molecular mechanisms causing these changes in barrier functions of the RPE, it was speculated that different expression levels of tight junction-related molecules or different levels of cell proliferation may be involved. As shown in Fig. 5, although qRT-PCR analysis of major tight junction-related molecules, ZO-1, Occludin, connexin 43 (Cx43) and Claudin-1, in the RPE showed that mRNA expression of these molecules fluctuated differently among conditions, those changes did not correspond to the changes of TEER values as shown above. In contrast, proliferation of ARPE19 cells was substantially increased under a high glucose condition compared to that under a low glucose condition, and proliferation of the cells under a high glucose condition was further enhanced by Ime or GW (Fig. 6), suggesting that the physical property of ARPE19 monolayers may be modulated by diverse cell proliferations affected by different glucose conditions, antidiabetic agents and/or GW.’.
- Very interestingly, the metabolic assays showed that Ime and Met induced an energetic shift towards glycolysis, while GW increased cellular respiration. This aspect is very interesting and could be explored to explain the side effects of Ime and Met in clinical studies.
Answer; Thank you very much for your valuable comment. As the reviewer pointed out, the findings in Figure 5 indicate that the metabolic activation by GW can be observed in the presence of Ime, but not Met, which is consistent with the report that Met shows uncompetitive inhibition of the respiratory chain, whereas Ime shows competitive inhibition of the respiratory chain (PMID: 33855213). These findings also reflect the fact that the risk of side effects of lactic acidosis is lower with Ime than that with Met (PMID: 35274817). In the revised manuscript, we have included these sentences in the 2nd paragraph of Discussion: ‘In contrast to such superiority of the effect of Met compared to that of Ime on mitochondrial respiration in hepatocyte function, our study showed that Ime and Met had similar effects in decreasing elevated levels of ROS (Fig. 2) and increasing TEER values (Fig. 3) of ARPE19 cells induced by high glucose, though the effects of Ime were more potent than those of Met. In addition to levels of ROS and TEER values, Ime, but not Met, had additional synergistical effects with the PPARa agonist GW on mitochondrial and glycolytic functions (Fig. 6). The results indicated that metabolic activation by GW can be observed in the presence of Ime, but not in the presence of Met, being consistent with the results of a previous study showing that Met shows uncompetitive inhibition of the respiratory chain, whereas Ime shows competitive inhibition of the respiratory chain [43]. Furthermore, these findings also reflect the fact that the risk of side effects of lactic acidosis is lower with Ime than that with Met [44].’.
- In order to be relevant, this MS should address different concentrations of the drugs as well as different time points, including long-term treatments. The questions generated by the experimental results should be discussed accordingly, and the cell toxicity effects should be carefully considered. These answers could indeed be important for the advancement of treatments for diabetes-related retinopathy.
Answer; We sincerely appreciate your excellent comment. I totally agree to address different concentrations of the drugs as well as different time points, including long-term treatments. In terms of different concentrations, other studies assessed similar ranging of Ime and Met from 0.25 to 10 mM using mouse and rat hepatocytes (PMID: 33855213, PMID: 36639407) and concentration dependent biological effects were shown. In addition, in those studies, much shorter exposure periods such as 0.5 to 12 hours of both drugs were used to evaluate biological effects. In addition, unlike in vivo experiments, long-term treatments are not practical due to drug-induced cytotoxic effects. In these situations, we carefully confirmed insignificant levels of drug induced cytotoxic effects by additional conditions: 2 mM and 5 mM concentration of Met and Ime during 24 hours through 4 days’ treatment. Therefore, these informations are included in the study limitation in Discussion:’ Fourthly, investigation of the effects of different concentrations of the drugs as well as different durations of treatment, including long-term treatment, is important for a better understanding of the efficacies of the drugs. Recently, it has been shown that AMPK activities in HepG2 cells and mouse primary hepatocytes were increased by Ime and Met in a dose-dependent manner with doses ranging from 0.25 to 10 mM [42], and 1 mM concentrations of both drugs were used to study glucose production, ATP/ADP ratio and mitochondrial functions in rat hepatocytes [43]. Even if different cell types were considered, these observations suggested that the concentration of 2 mM of Ime and Met used in our study may be a suitable experimental condition for evaluating the effects of the drugs. Furthermore, in those studies, the effects of the drugs were evaluated in much shorted exposure time periods such as 0.5 to 12 hours [42] than the exposure period of 24 hours in our study. In contrast to in vivo experiments using animal models, since the cell toxicity effects should be carefully considered in the case of in vitro cell culture experiments, insignificant levels of cytotoxic effects by both drugs until 24 hours were confirmed as shown Fig. 1. Therefore, to overcome these study limitations, additional study is required to determine the pharmacological effects of Ime on the pathogenesis of DR using additional intraocularly derived cells including primary cultured RPE cells and in vivo animal models as well as using clinically used PPARa agonists in our next projects.’.
Reviewer 3 Report
Comments and Suggestions for Authors
The authors submitted an interesting manuscript regarding the effect of the combination of the PPAR agonist GW7646 and imeglimin in the pathophysiology of diabetic retinopathy.
The manuscript is well written and organized although several key answers were not explained. But the manuscript suggest that future work will help understand the role of these drugs in the development of the disease.
Suggestions/Comments:
Line 90 - the sentence is a bit confusing. I suggest rewriting it.
Author Response
Dear Editor,
Thank you very much for the constructive comments concerning our manuscript “Combination of the PPARα agonist GW7646 and imeglimin has potent effects on high glucose-induced cellular biological responses in human retinal pigment epithelium cells.”. We carefully checked all of the Editor and reviewers’ comments and prepared a revised version of our paper that takes these comments into account. The changes are listed below.
Editors’ comment
As pointed out, plagiarism is reduced.
Reviewer 3 comments
The authors submitted an interesting manuscript regarding the effect of the combination of the PPAR agonist GW7646 and imeglimin in the pathophysiology of diabetic retinopathy. The manuscript is well written and organized although several key answers were not explained. But the manuscript suggest that future work will help understand the role of these drugs in the development of the disease.
Answer; We sincerely appreciate your excellent comment. As suggested, in addition to study limitation related to several key answers currently unidentified, the conclusion on future perspectives of hyperglycemia-induced mitochondrial dysfunction on studies outcome considering production of ROS is included:’ Ime showed stronger effects than those of Met, along with additional synergistical effects with GW, in reducing oxidative stress and improving barrier function under a high glucose condition in ARPE19 cells, suggesting the potential of Ime for inhibiting DR progression. These pharmacological effects were observed even under a high glucose condition, suggesting that they can be independent of glycemic management in patients with DM. Clinical trials are needed to verify the efficacy of Ime against DR at various stages in the future.’.
Round 2
Reviewer 1 Report
Comments and Suggestions for Authors
The authors have addressed all my concerns but the issue related to plagiarism in methodology must be addressed before publication
Author Response
Dear Editor,
Thank you very much for the constructive comments concerning our manuscript “Combination of the PPARα agonist GW7646 and imeglimin has potent effects on high glucose-induced cellular biological responses in human retinal pigment epithelium cells.”. We carefully checked all of the Editor and reviewers’ comments and prepared a revised version of our paper that takes these comments into account. The changes are listed below.
Reviewer 1 comments
The authors have addressed all my concerns but the issue related to plagiarism in methodology must be addressed before publication.
Answer; We sincerely appreciate your excellent comment. As pointed out, the issue related to plagiarism in methodology is corrected.
Reviewer 2 Report
Comments and Suggestions for Authors
Unfortunately, the revised version did not address my previous concerns about the proposed substances' side effects and therapeutic benefits.
The most important ones imply the toxicity of the tested substances at the investigated concentrations. This toxic effect was shown to start on day 2 of treatment of the cell cultures with the proposed concentrations (Fig 1). However, these harmful effects are not considered in the revised version.
The authors did not address my concern regarding decreased ROS after treatments, which were under low glucose condition values. The therapeutic benefits versus side effects are again not balanced. "While all the treatments lead to a significant decrease in ROS, these levels were much lower than in low glucose conditions, which challenges their therapeutic benefit".
The only consideration is in the discussion section, but not in the experimental approach related to this study: "Furthermore, these findings also reflect the fact that the risk of side effects of lactic acidosis is lower with Ime than that with Met [44]".
Author Response
Dear Editor,
Thank you very much for the constructive comments concerning our manuscript “Combination of the PPARα agonist GW7646 and imeglimin has potent effects on high glucose-induced cellular biological responses in human retinal pigment epithelium cells.”. We carefully checked all of the Editor and reviewers’ comments and prepared a revised version of our paper that takes these comments into account. The changes are listed below.
Reviewer 2 comments
Unfortunately, the revised version did not address my previous concerns about the proposed substances' side effects and therapeutic benefits.
- The most important ones imply the toxicity of the tested substances at the investigated concentrations. This toxic effect was shown to start on day 2 of treatment of the cell cultures with the proposed concentrations (Fig 1). However, these harmful effects are not considered in the revised version.
Answer; We sincerely appreciate your excellent comment. As pointed out, more detail including harmful effects in Fig. 1 is included in the corresponding result: ‘As shown in Fig. 1, 1) no significant cytotoxic effects of Met or Ime were detected in the concentration ranges between 0 and 2 mM at day 1 under low glucose and high glucose conditions (panel A), and 2) toxic effect was observed at 5 mM concentration of these reagents at day 1 and more than two days’ treatment of the cell cultures at 2 mM of these reagents under a high glucose condition (panel B).’.
- The authors did not address my concern regarding decreased ROS after treatments, which were under low glucose condition values. The therapeutic benefits versus side effects are again not balanced. "While all the treatments lead to a significant decrease in ROS, these levels were much lower than in low glucose conditions, which challenges their therapeutic benefit".
Answer; We sincerely appreciate your excellent comment. As suggested, this information is included in the study limitation in Discussion: ‘Fifthly, while the mono-treatment of Ime and treatments of Met or Ime with GW lead to a significant decrease in ROS under a high glucose condition, these levels were much lower than in low glucose conditions as shown in Fig. 2. Although biological significance of such decreased levels of ROS after the treatments, which were under levels of ROS at physiological low glucose condition (5.5 mM) remains to be elucidated, this may bring therapeutic benefits on diabetic and none-diabetic RPE.’.
- The only consideration is in the discussion section, but not in the experimental approach related to this study: "Furthermore, these findings also reflect the fact that the risk of side effects of lactic acidosis is lower with Ime than that with Met [44]".
Answer; We sincerely appreciate your excellent comment. As pointed out, I agree that this information is not related to the experimental approach related to this study, and therefore, this is removed.
Round 3
Reviewer 2 Report
Comments and Suggestions for Authors
Unfortunately, the revised version did not address my previous concerns about the proposed substances' side effects and therapeutic benefits in longer-term treatments.
Author Response
Dear Reviewer 2,
We would like to thank you for taking time out of your busy schedule to review our manuscript “Combination of the PPARα agonist GW7646 and imeglimin has potent effects on high glucose-induced cellular biological responses in human retinal pigment epithelium cells.” with the Editor’s decision of minor revision.
We have carefully read your comments throughout the peer review process. As far as we interpret, the gap between low levels of ROS production and decreased cell viability induced by Met or Ime may remain your major concern. Thus, we add additional paragraph of possible interpretation in the Discussion section of the revised manuscript with green markers: ‘Since insignificant long-term side effects of Met and Ime in clinical settings have been confirmed, these agents are already used as common therapy for patients with hyperglycemia [14,15,25,26]. However, in the present in vitro study, treatment with 2 mM Met and 2 mM Ime did not affect the cellular viability of ARPE19 cells for up to 24 hrs, whereas more than 2 days culture with these agents slightly but significantly de-creased the cellular viability without additional effects up to 5 days (Fig. 1). These findings may appear to contradict the effect of Met and Ime on reducing ROS produc-tion (Fig. 2). Considering the results of the Seahorse analysis showing that 2 mM Met and 2 mM Ime shifted the cellular metabolism of ARPE19 from mitochondrial respira-tion to glycolysis (Fig. 7), and the fact that retinal pigment epithelial cells are vulnera-ble to hypoxia or nutrition deprivation, the effects on cell viability by Met and Ime are presumably due to cell metabolic deprivation rather than ROS-dependent cellular damage or cell death. From this point of view, the fact that the metabolic activation by GW was clearly observed when it was used in combination with Ime, but not with Met, supports the notion that Met and Ime differently affect cellular biological properties including redox status and cellular metabolism. Collective these findings suggested that the effect of Ime induced reduction of levels of ROS on cellular metabolism may be minimized by using Ime in combination with PPARα agonists such as GW. To confirm our speculation, further studies using in vivo animal models or clinical studies focusing on the long-term effects on DR in Met and Ime in combination with PPARα agonists are warranted.’.
We hope that this revision is sufficient and the manuscript is ready for publication in Bioengineering.
Sincerely yours,
Tatsuya Sato and Hiroshi Ohguro, co-corresponding authors of this manuscript